# Geometric Stability of Representation Manifolds as a Training-Free Diagnostic for Studying Data Augmentations

**Ahmad Taha**
Research Center of the Artificial Intelligence Institute
Innopolis, Russia
`a.taha@innopolis.university`

**Rustam Lukmanov**
Research Center of the Artificial Intelligence Institute
Innopolis, Russia
`r.lukmanov@innopolis.ru`

## Abstract

Data augmentation is the primary mechanism for defining representation invariances in self-supervised learning (SSL), but the selection of augmentations remains largely empirical and computationally costly, as it typically requires repeated full training runs for validation. We introduce a training-free diagnostic that evaluates augmentations based on the geometric stability of the learned embedding manifold. Our method uses Procrustes analysis to measure the non-rigid distortions caused by augmentation operators in the feature space of a strong pre-trained encoder. We observe a statistically significant relationship between geometric preservation and the semantic consistency of representations in high-dimensional space. These findings establish global geometric stability as a computationally efficient, training-free diagnostic for studying the semantic effects of data augmentations. Furthermore, we investigate the boundary conditions by analyzing situations in which geometric proximity decouples from instance-level discriminability. Our framework provides a principled and mathematically grounded approach for evaluating augmentations in medical and general-purpose foundation models.

## 1 Introduction

Self-supervised learning (SSL) relies on data augmentation to define invariances in learned representations (Cosentino et al., 2022; Morningstar et al., 2024). In medical imaging the labeled data is scarce and the distribution shifts are common (Huang et al., 2023) hence selecting augmentations is critical (Azizi et al., 2023). However, augmentation selection remains largely empirical and computationally expensive, often requiring full training cycles to validate choices (Cubuk et al., 2019).

**Proposed Diagnostic.** We introduce a training-free diagnostic based on geometric stability of a frozen encoders embedding manifold. The diagnostic computes a scalar distortion score that measures the non-rigid deformation an augmentation induces in feature space where low distortion correlates with preserved instance identity while large distortion signals potential semantic corruption. We test our hypothesis by using a powerful, pre-trained foundation model. The core assumption is that such an encoder has already learned a rich set of visual invariances, making its feature manifold a high-quality proxy for what a new model would learn. By probing this manifold, we can efficiently study the effects of augmentations without the cost of repeated training cycles (Nayal et al., 2025).

**Approach.** We propose a diagnostic framework that quantifies the stability of the feature manifold via Procrustes analysis (Goodall, 1991). To validate this geometric metric, we contrast it against a semantic integrity score measured via zero-shot contrastive discriminability. Analyzing 18 distinct augmentations on endoscopic video data (HyperKvasir (Borgli et al., 2020)) and we report two key findings: (1) we observe a statistically significant negative correlation (Spearman $\rho \approx -0.67$, $p < 0.01$) between geometric distortion and semantic preservation, validating geometric stability as a utility proxy, (2) we identify specific divergence regimes where geometric proximity fails to guarantee semantic identity.

**Contribution.** Our work makes a twofold contribution. First, we introduce and validate a novel, training-free diagnostic for measuring the geometric impact of augmentations. Second, we use this

tool to empirically identify a strong predictive relationship between geometric stability and semantic integrity. We then investigate the boundary conditions of this, identifying failure modes that reveal deeper insights into the non-geometric priors learned by encoders.

## 2 METHOD

### 2.1 NOTATION AND SETUP

Let $f : \mathbb{R}^{H \times W \times 3} \to \mathbb{R}^d$ be a frozen, pre-trained encoder (PE-Core-L14-336, $d = 1024$) (Bolya et al., 2025) mapping input frames to $\ell_2$-normalized feature vectors. For a batch of $N$ video clips, we compute the feature matrices:

$$\mathbf{Z} = f(\mathbf{X}) \in \mathbb{R}^{N \times d} \quad \text{(original manifold)}, \tag{1}$$

$$\mathbf{Z}' = f\big(T(\mathbf{X})\big) \in \mathbb{R}^{N \times d} \quad \text{(augmented manifold)}, \tag{2}$$

where $T \in \mathcal{T}$ represents a specific augmentation function.

### 2.2 GEOMETRIC INTEGRITY: PROCRUSTES DISTANCE

To quantify how much $T$ distorts the relative geometry of the data cloud we use Procrustes Analysis, which is invariant to global rotation, translation, and scaling. Let $\mathbf{Z}_c$ and $\mathbf{Z}'_c$ denote the column-centered feature matrices. We seek the orthogonal matrix $\mathbf{Q} \in \mathrm{O}(d)$ that best aligns the augmented cloud to the original:

$$\mathcal{D}_{\text{geo}}(\mathbf{Z}, \mathbf{Z}') \;=\; \min_{\mathbf{Q} \in \mathrm{O}(d)} \big\| \mathbf{Z}_c - \mathbf{Z}'_c \mathbf{Q} \big\|_F^2. \tag{3}$$

The optimal orthogonal matrix $\mathbf{Q}$ is obtained via singular value decomposition of the cross-covariance:

$$(\mathbf{Z}'_c)^\top \mathbf{Z}_c = \mathbf{U} \boldsymbol{\Sigma} \mathbf{V}^\top, \qquad \mathbf{Q} = \mathbf{V} \mathbf{U}^\top. \tag{4}$$

This residual $\mathcal{D}_{\text{geo}}$ quantifies the non-rigid distortion introduced by the augmentation. This score is intended as a diagnostic signal for augmentation evaluation (not as a training objective) as it enables screening and hypothesis-driven experiments about augmentation effects without model fine-tuning or labeled data.

### 2.3 SEMANTIC INTEGRITY: CONTRASTIVE AUROC

To measure whether the augmentation preserves the identity of an instance so it remains distinguishable from other, we define a zero-shot retrieval metric. For each sample $i$ define the positive pair $(\mathbf{z}_i, \mathbf{z}'_i)$ and the set of negative pairs $\{(\mathbf{z}_i, \mathbf{z}_j)\}_{j \neq i}$. We compute the distributions of cosine similarities:

$$S_{\text{pos}} = \{ \mathbf{z}_i^\top \mathbf{z}'_i : \; i = 1, \ldots, N \}, \qquad S_{\text{neg}} = \{ \mathbf{z}_i^\top \mathbf{z}_j : \; i = 1, \ldots, N, \; j \neq i \}. \tag{5}$$

The Semantic Integrity is defined as the area under the ROC curve (AUROC), the probability that a positive pair scores higher than a negative pair:

$$\mathcal{S}_{\text{sem}} \;=\; \mathbb{P}\big(s_{\text{pos}} > s_{\text{neg}}\big), \qquad s_{\text{pos}} \sim S_{\text{pos}}, \; s_{\text{neg}} \sim S_{\text{neg}}. \tag{6}$$

We note that $\mathcal{S}_{\text{sem}}$ measures *instance-level* discrimination: a positive pair consists of the same frame under augmentation, not two frames from the same class. We therefore use semantic integrity to mean preservation of instance identity in feature space, since this is what contrastive SSL objectives directly optimize. A class-level version, where positives come from the same class instead of the same instance, is a natural direction for future work.

Each augmentation is evaluated using one random sample per frame. This isolates the geometric effect of a single realization of $T$, but does not capture the variability across different stochastic views during training. Averaging $\mathcal{D}_{\text{geo}}$ over multiple samples per frame would naturally extend the analysis from a single draw to the full augmentation distribution.

**Experimental Setup.** We evaluate 18 transformations categorized by type (geometric, photometric, and corruption) on $N = 250$ video from the HyperKvasir dataset (Borgli et al., 2020). This subset covers 5 classes (polyps, normal, inflammation, bleeding, and tools) to ensure structural diversity. A comprehensive list of all transformations and their parameters is provided in Appendix A.1.

## 3 Results

### 3.1 Qualitative Manifold Analysis

To investigate the topological impact of augmentations we first visualize the embedding space geometry. Figure 1 displays the projection of feature clusters for selected transformations. Effective augmentations (for example, `color_jitter`, `crop`) maintain the separability of class clusters even when inducing significant displacement in the feature space. In contrast transformations such as `invert` scramble local neighborhoods, collapsing distinct clusters into inseparable regions despite retaining the global shape of the point cloud.

### 3.2 A Strong Geometric-Semantic Relationship

We quantify these observations across $K = 18$ augmentations. As shown in Figure 2, we identify a statistically significant negative correlation (Spearman $\rho = -0.67$, $p = 0.003$) in the high-dimensional space between Geometric Distortion ($\mathcal{D}_{geo}$) and Semantic Integrity ($\mathcal{S}_{sem}$).

This strong statistical relationship validates our hypothesis: for the majority of transformations, geometric stability is a significant predictor of an augmentation's utility. Mild geometric perturbations (for example, `perspective`) cluster in the High-Fidelity Regime ($\mathcal{D}_{geo} < 0.03, \mathcal{S}_{sem} > 0.98$), while aggressive corruptions (for example, `noise_aggressive`) degrade both metrics.

### 3.3 Regimes of Geometric-Semantic Divergence

Despite the general trend, we identify a critical regime where geometric proximity is not correlated with semantic identity (Table 1).

For example, `invert` induces minimal geometric shift ($\mathcal{D} = 0.077$), ranking 5th in geometric stability but it causes semantic failure ($\mathcal{S} = 0.155$). This indicates that the encoder relies on specific appearance priors that are orthogonal to the primary geometric axes of the manifold.

Table 1: Augmentations with low geometric distortion ($\mathcal{D}_{geo}$) but high semantic failure (Low $\mathcal{S}_{sem}$).

| Augmentation | $\mathcal{D}_{geo}$ (Lower is better) | $\mathcal{S}_{sem}$ (Higher is better) | Geo Rank |
|---|---|---|---|
| `perspective` | 0.014 | 0.994 | 1st (Reference) |
| `invert` | 0.077 | 0.155 | 5th (Diverged) |
| `motion_blur` | 0.075 | 0.101 | 10th (Diverged) |

## 4 Discussion

### 4.1 Manifold Topology as a Proxy for Utility

Our diagnostic has implications for automated augmentation search such as AutoAugment (Cubuk et al., 2019): candidate augmentations can be screened via Procrustes analysis before training, pruning those that induce manifold collapse or extreme geometric distortion to reduce the search space.

### 4.2 Generalization to Non-Contrastive Frameworks

We employ a contrastive metric (AUROC) for validation, our geometric diagnostic is loss-agnostic as it measures properties of the data manifold itself. We hypothesize that the requirement for manifold stability applies equally to non-contrastive architectures, such as self-distillation (DINO (Caron et al., 2021), BYOL (Grill et al., 2020)) or masked image modeling (He et al., 2022). Quantifying the specific sensitivity of these objectives to geometric distortions remains as future investigation.

### 4.3 Towards Downstream Validation

The natural next step is to connect the diagnostic to downstream SSL performance: pretrain a contrastive or self-distillation model (e.g., SimCLR or DINO) with each candidate augmentation on

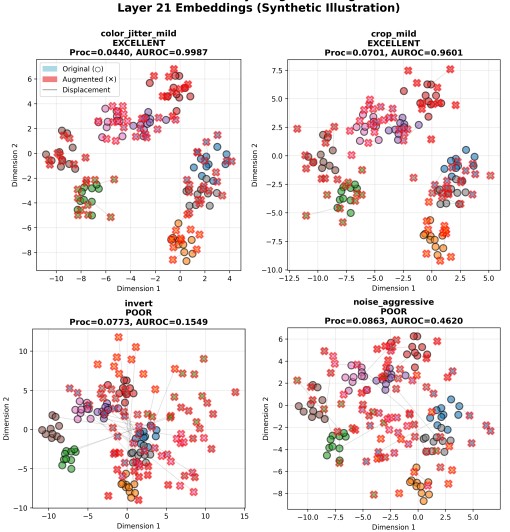

Figure 1: **Manifold Topology.** t-SNE showing that effective augmentations maintain cluster separation, while pathological ones scramble neighborhoods. *t-SNE is used for qualitative visualization, all quantitative metrics use original space.*

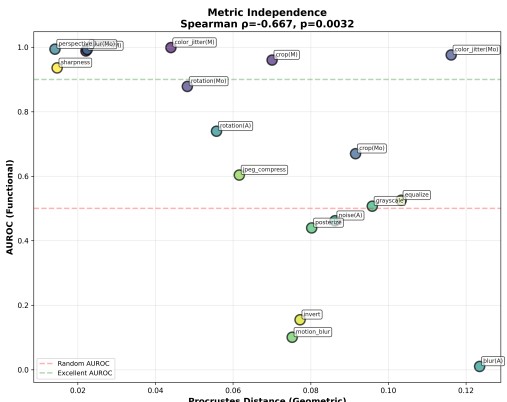

Figure 2: **The Correlation.** We observe a strong negative correlation ($\rho = -0.67$) between Geometric Distortion and Semantic Integrity, with clear outliers.

an SSL-feasible subset, then correlate $\mathcal{D}_{\text{geo}}$ (from the frozen reference encoder) with linear-probe accuracy across the 18 augmentations. A strong correlation would establish the diagnostic as a screening tool where a weak one would localize where frozen-encoder geometry diverges from biases acquired during SSL training, which is itself informative. We treat this as future work since it requires the SSL training cycles our diagnostic was designed to avoid.

## 4.4 LIMITATIONS

Our main limitation, and a key assumption of the method, is that the analysis depends on the feature space of a single encoder (PE-Core-L14-336). Encoders trained with different objectives or augmentation strategies learn different invariances, so the exact geometric-semantic relationship may change across models. In particular, methods such as MAE, DINO, and SimCLR learn different priors, meaning the absolute correlation values reported here may not directly transfer between encoders. The geometric stability framework itself remains applicable, but the threshold for what should be considered low or high distortion is encoder-dependent. We choose PE-Core-L14-336 because it has demonstrated state-of-the-art performance on HyperKvasir (Taha and Lukmanov, 2025b;a) and provides a high-quality proxy manifold. Future work should examine how these geometric-semantic relationships generalize across architectures (e.g., ConvNeXt (Liu et al., 2022)) and SSL objectives, and characterize the stochastic structure of $\mathcal{D}_{\text{geo}}$ under repeated sampling.

## 5 CONCLUSION

We introduce a training-free diagnostic for augmentation utility based on geometric stability of the embedding manifold, quantifying non-rigid distortion via Procrustes analysis. We establish a strong empirical relationship ($\rho \approx -0.67$) connecting geometry to instance-level semantics, providing practitioners with a principled tool to screen augmentations before training begins.

## ACKNOWLEDGMENTS

The study was supported by the Ministry of Economic Development of the Russian Federation (agreement No. 139-10-2025-034 dd. 19.06.2025, IGK 000000C313925P4D0002).

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

# A  APPENDIX

## A.1  LIST OF EVALUATED AUGMENTATIONS

We evaluate 18 transformations (Ramesh et al., 2023) across varying intensities. Special augmentations refer to operations that do not fit strictly into geometric, photometric, or corruption categories like sharpness adjustments or histogram equalization.

- *Geometric:* Rotation, Crop, Perspective.
- *Photometric:* Color Jitter, Grayscale, Invert, Equalize.
- *Corruptions:* Gaussian Blur, Gaussian Noise, JPEG Compression, Motion Blur.

## A.2  ADDITIONAL STATISTICAL ANALYSIS

### A.2.1  CORRELATION ROBUSTNESS AND OUTLIER ANALYSIS

To confirm the robustness of our reported correlation ($\rho = -0.67$, $p = 0.003$), we perform bootstrap resampling and leverage analysis. A bootstrap over 10,000 resamples yields a 95% confidence interval of $[-0.88, -0.20]$, confirming statistical significance across the full range of resampled distributions.

We also compute Cook's distance  (Cook, 1979) to assess whether any single augmentation disproportionately drives the correlation. We found that no augmentation exceeds the standard influence threshold (Cook's $D > 1.0$; maximum observed: $D = 0.57$ for `color_jitter_moderate`). This indicates that the geometric-semantic deviations we observe are systematic and not driven by individual outliers. Removing the three highest-leverage points (`color_jitter_moderate`, `blur_aggressive`, `motion_blur`) *strengthens* the correlation to $\rho = -0.77$ ($p = 0.0008$), this confirm these points represent transitions and not noise.

Table 2: Cook's distance analysis. No point exceeds the influence threshold ($D > 1$), confirming deviations are systematic.

| Augmentation | Cook's $D$ | $\mathcal{D}_{\text{geo}}$ | $\mathcal{S}_{\text{sem}}$ |
|---|---|---|---|
| color_jitter_moderate | 0.572 | 0.116 | 0.976 |
| blur_aggressive | 0.183 | 0.124 | 0.010 |
| motion_blur | 0.114 | 0.075 | 0.101 |
| invert | 0.089 | 0.077 | 0.155 |
| crop_mild | 0.041 | 0.070 | 0.960 |

### A.2.2  CLASS-CONDITIONAL SEMANTIC INTEGRITY

We compute AUROC conditioned on each of the five HyperKvasir classes to reveal which augmentations disproportionately harm specific categories (Table 3).

First, Polyps is the most fragile class: it got affected the most by `invert` (0.014), `motion_blur` (0.000) and `noise_aggressive` (0.096) (which is expected). This is because polyps being defined by textural and color signatures, fine vascular patterns and irregularities in mucosa which are destroyed by any augmentation that alter high-frequency content or color priors.

Table 3: Per-class AUROC. Polyps is systematically the most vulnerable class across destructive augmentations. `crop` disproportionately damages Inflammation.

| Augmentation | Polyps | Normal | Inflam. | Bleed. | Tools |
|---|---|---|---|---|---|
| color_jitter_mild | 0.999 | 1.000 | 0.998 | 1.000 | 1.000 |
| perspective | 1.000 | 0.991 | 0.989 | 0.985 | 0.974 |
| crop_mild | 1.000 | 1.000 | 0.822 | 0.993 | 0.998 |
| crop_moderate | 0.453 | 0.623 | **0.448** | 0.766 | 0.686 |
| blur_moderate | 0.961 | 0.997 | 0.984 | 0.986 | 0.986 |
| invert | **0.014** | 0.027 | 0.108 | 0.330 | 0.289 |
| motion_blur | **0.000** | 0.103 | 0.066 | 0.219 | 0.219 |
| blur_aggressive | 0.000 | 0.000 | 0.007 | 0.036 | 0.005 |
| noise_aggressive | **0.096** | 0.377 | 0.335 | 0.614 | 0.556 |

Second, `crop_mild` preserves most classes (AUROC > 0.99), but degrades Inflammation to 0.822. At moderate intensity, both Polyps (0.453) and Inflammation (0.448) collapse to value close to random or even less, but Bleeding gain some discriminability (0.766). This suggests that inflammatory lesions, like polyps are defined by local textural detail easily blocked by cropping but bleeding mostly by global color shift which is more robust to spatial occlusion.

### A.3   HYPERKVASIR DATA CHARACTERISTICS

The HyperKvasir dataset  (Borgli et al., 2020) presents a significant challenge due to high intra-class and inter-class similarity. Figure 3 illustrates this. Frames from the same video (row-wise) have high similarities changes due to camera motion and lighting. Frames from different videos (column-wise) even across classes can appear visually similar.

Table 4 presents the complete evaluation of all 18 augmentations.

Table 4: Complete results for all evaluated augmentations, ranked by AUROC.

| Augmentation | $\mathcal{D}_{geo}$ | $\mathcal{S}_{sem}$ | Category |
|---|---|---|---|
| color_jitter_mild | 0.044 | 0.999 | Photometric |
| perspective | 0.014 | 0.994 | Geometric |
| blur_moderate | 0.022 | 0.993 | Corruption |
| rotation_mild | 0.022 | 0.988 | Geometric |
| color_jitter_moderate | 0.116 | 0.976 | Photometric |
| crop_mild | 0.070 | 0.960 | Geometric |
| sharpness | 0.015 | 0.936 | Special |
| rotation_moderate | 0.048 | 0.878 | Geometric |
| rotation_aggressive | 0.056 | 0.739 | Geometric |
| crop_moderate | 0.092 | 0.670 | Geometric |
| jpeg_compress | 0.062 | 0.604 | Corruption |
| grayscale | 0.096 | 0.507 | Photometric |
| noise_aggressive | 0.086 | 0.462 | Corruption |
| posterize | 0.080 | 0.440 | Special |
| equalize | 0.103 | 0.381 | Special |
| invert | 0.077 | 0.155 | Special |
| motion_blur | 0.075 | 0.101 | Corruption |
| blur_aggressive | 0.124 | 0.010 | Corruption |

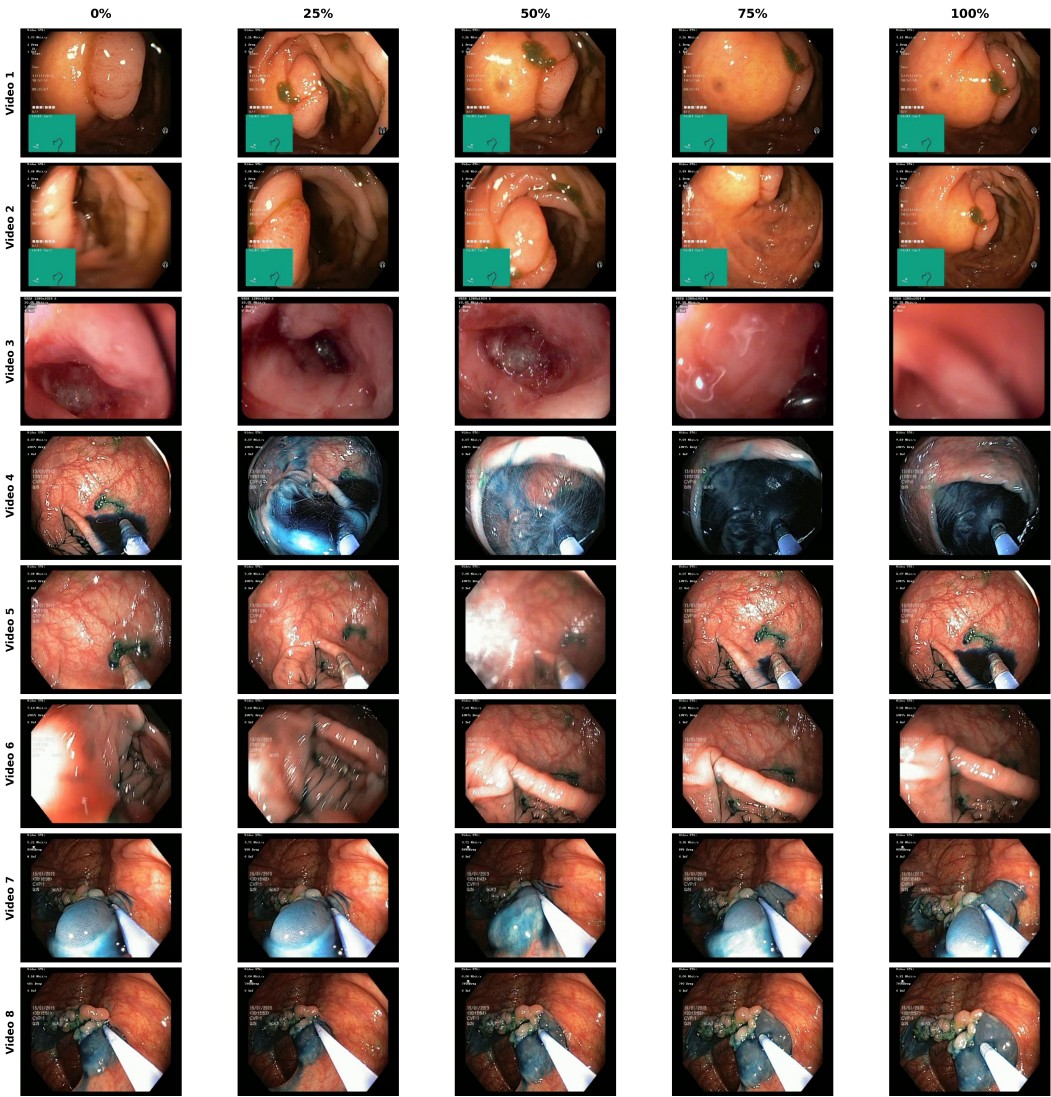

Figure 3: **The HyperKvasir Challenge.** Frames sampled at different time steps (0% to 100%) from nine distinct videos. Note the high visual similarity between frames from different videos, complicating instance discrimination.

