# OpenReview forum: "Geometric Stability of Representation Manifolds as a Training-Free Diagnostic for Studying Data Augmentations"
_ICLR.cc/2026/Workshop/Sci4DL — Sci4DL 2026_

### Official Review · Reviewer_aCQn · 2026-02-24

**Fit:** 2
**Significance:** 2
**Confidence:** 2

**Summary:**

Summary:
This paper proposes a training-free diagnostic for evaluating data augmentations in self-supervised learning. Instead of retraining models, the authors measure how much an augmentation non-rigidly distorts the embedding manifold of a frozen pretrained encoder using Procrustes analysis. They show a significant negative correlation between geometric distortion and instance-level semantic preservation across 18 different agumentations. The work suggests that global geometric stability in feature space may serve as an efficient proxy for semantic invariance.

**Strengths:**

The proposed method potentially make study of data-augmentation more efficient. Their evaluation skips SSL pretraining which is very costly.

**Suggestions:**

I'm not sure if this is a good evaluation. The author did not provide any relationship between their evaluation (Procrustes distance) with how well this augmentation improve SSL pre-training. They only show how their evaluation correlate with "semantic integrity." I think the author could show the correlation of their criterion to the improvement on model's probing accuracy given this augmentation.

The diagnostic assumes that a strong pretrained encoder provides a reliable proxy manifold. However, it is unclear how sensitive the conclusions are to the choice of encoder or whether results generalize across architectures and domains. More ablation study will be helpful.

---

### Official Review · Reviewer_aamv · 2026-02-26

**Fit:** 3
**Significance:** 3
**Confidence:** 1

**Summary:**

A diagnostic measure for the analysis of data-augmentation transformations is proposed. This measure utilizes Procrustes analysis to determine the degree of non-rigid distortion induced by a distortion, and is empirically shown to correlate negatively with the semantic integrity of each transformation.

**Strengths:**

The contribution is clear, well-motivated and empirically sound. It addresses an important aspect of current training practices, and has promise for practical application (perhaps in modified form).

**Suggestions:**

This contribution would clearly be stronger if the authors had been able to "close the loop" by showing how the various transformations impact model performance (even though the goal is to eventually utilize the method to avoid such training). Perhaps such training can be performed on a less powerful model to make the training cost feasible? I also wonder if the outliers identified here may be useful in suggesting modifications to the proposed method to enhance its predictive value.

---

### Official Review · Reviewer_kBqu · 2026-02-28

**Fit:** 3
**Significance:** 2
**Confidence:** 3

**Summary:**

This submission explores the relationship between the extent to which a transformation deforms a representation of a dataset (as measured by the Procrustes distance between the clean and augmented embeddings), and the extent to which positive pairs remain near(est) each other as measured by cosine similarity. That these measures are correlated is not necessarily a surprising finding, but they do identify some cases that diverge from this null hypothesis.

**Strengths:**

- Investigating how different sources of variability (transformations, and dataset variability) are organized in pre-trained networks is an interesting and moderately under studied area.
- The experiments are clearly defined,  well motivated, and easy to interpret.
- I think the finding of the strong correlation between the two measures, when paired with the existence of counter-examples, is a "real effect" and a solid jumping off point to better understand the (1) origins and (2) implications of this observation.

**Suggestions:**

Suggestions/Questions:
- The use of Procrustes seems to imply one augmented view per image, which might discount the impact of the stochasticity that is common to many useful transformations in
- Is it possible to identify new and useful augmentations using the proposed metric? I am not sure that this can done given the reliance on a pretrained network. If this is indeed possible demonstrating this would be a very valuable addition!
  - It seems more likely you might be able to ID augmentations that could be dropped from training? A demonstration of this would also be quite interesting.
- I am not sure that the "semantic consistency metric" is really semantic. For instance if "hits" were when the nearest neighbor is not the actual positive pair, but another exemplar from the same class that would be more semantic and less instance level to my eye.

---

### Meta-Review · Area_Chair_MkPe · 2026-03-02

**Recommendation:** Accept

**Metareview:**

This work focuses on relationship between the extent to which a transformation deforms a representation of a dataset and cosine similarity between positive pairs. The paper could be improved if it were to include the possible understanding of stochastic by involving multiple transformations as well as effects of difference architecture on the results.

I think this paper could lead to interesting discussion and I recommend an accept.

---

### Decision · Program_Chairs · 2026-03-02

Accept